# Preventing COVID-19 spread in closed facilities by regular testing of employees—An efficient intervention in long-term care facilities and prisons?

Henri Christian Junior Tsoungui Obama[1☯], Nessma Adil Mahmoud Yousif[1☯], Looli Alawam Nemer[2], Pierre Marie Ngougoue Ngougoue[1], Gideon Akumah Ngwa[1,3], Miranda Teboh-Ewungkem[4], Kristan Alexander Schneider[1☯]*

1 Department of Applied Computer- and Biosciences, University of Applied Sciences Mittweida, Mittweida, Germany, 2 African Institute for Mathematical Sciences Cameroon, Limbe, Cameroon, 3 Department of Mathematics, University of Buea, Buea, Cameroon, 4 Department of Mathematics, Lehigh University, Bethlehem, Pennsylvania, United States of America

☯ These authors contributed equally to this work.
* kristan.schneider@hs-mittweida.de

## Abstract

### Background

Different levels of control measures were introduced to contain the global COVID-19 pandemic, many of which have been controversial, particularly the comprehensive use of diagnostic tests. Regular testing of high-risk individuals (pre-existing conditions, older than 60 years of age) has been suggested by public health authorities. The WHO suggested the use of routine screening of residents, employees, and visitors of long-term care facilities (LTCF) to protect the resident risk group. Similar suggestions have been made by the WHO for other closed facilities including incarceration facilities (e.g., prisons or jails), wherein parts of the U.S., accelerated release of approved inmates is taken as a measure to mitigate COVID-19.

### Methods and findings

Here, the simulation model underlying the pandemic preparedness tool CovidSim 1.1 (http://covidsim.eu/) is extended to investigate the effect of regularly testing of employees to protect immobile resident risk groups in closed facilities. The reduction in the number of infections and deaths within the risk group is investigated. Our simulations are adjusted to reflect the situation of LTCFs in Germany, and incarceration facilities in the U.S.

COVID-19 spreads in closed facilities due to contact with infected employees even under strict confinement of visitors in a pandemic scenario without targeted protective measures. Testing is only effective in conjunction with targeted contact reduction between the closed facility and the outside world—and will be most inefficient under strategies aiming for herd immunity. The frequency of testing, the quality of tests, and the waiting time for obtaining test results have noticeable effects. The exact reduction in the number of cases depends on

**Data Availability Statement:** Only simulated data is used. The programming code is available on GitHub: http://doi.org/10.5281/zenodo.4634314.

**Funding:** K. A. S. could not have put together the research team without the supported by the German Academic Exchange (DAAD; https://www.daad.de/de/; Project-ID 57417782), the Sächsisches Staatsministerium für Wissenschaft, Kultur und Tourismus and Sächsische Aufbaubank – Förderbank (SMWK-SAB; https://www.smwk.sachsen.de/; https://www.sab.sachsen.de/; project "Innovationsvorhaben zur Profilschärfung an Hochschulen für angewandte Wissenschaften", Project-ID 100257255), the Federal Ministry of Education and Research (BMBF) and the DLR (Project-ID 01DQ20002; https://www.bmbf.de/; https://www.dlr.de/). G. N. is supported by the German Academic Exchange (DAAD; https://www.daad.de/de/; Project-ID 57479556). The funders had no role in study design, data collection and analysis, decision to publish, or preparation of the manuscript.

**Competing interests:** The authors have declared that no competing interests exist.

disease prevalence in the population and the levels of contact reductions. Testing every 5 days with a good quality test and a processing time of 24 hours can lead up to a 40% reduction in the number of infections. However, the effects of testing vary substantially among types of closed facilities and can even be counterproductive in U.S. IFs.

## Conclusions

The introduction of COVID-19 in closed facilities is unavoidable without a thorough screening of persons that can introduce the disease into the facility. Regular testing of employees in closed facilities can contribute to reducing the number of infections there, but is only meaningful as an accompanying measure, whose economic benefit needs to be assessed carefully.

## Introduction

The global COVID-19 pandemic that emerged in Wuhan, China in December 2019 was declared a Public Health Emergency of International Concern by the WHO Director-General in late January 2020 and drastically changed the way of living across the globe [1]. SARS-CoV-2 is an extremely contagious virus affecting the respiratory system [2]. While most infections are asymptomatic and mild, severe infections are life-threatening [3–6]. If the virus affects the lung it can result in diffuse pneumonia, requiring oxygen supply, hospital, or even ICU treatment [7–11]. With no effective treatment against the virus, severe episodes can result in death by multiple organ failure [12]. Moreover, severe (and even mild) infections can cause substantial long-term effects, potentially imposing long-term burdens on healthcare systems [13, 14]. From the beginning of the pandemic, older adults and individuals with underlying medical conditions, particularly lung or heart disease, diabetes, obesity, etc. are associated with an increased risk of developing serious complications from SARS-CoV-2 infections [15]. The Centers for Disease Control and Prevention (CDC) identified people aged 65 years and older and people living in a long-term care facility (LTCF) as high-risk groups.

Different control measures, were implemented by governments across the globe to prevent the spread of the pandemic, including social distancing (cancellation of mass gatherings and events, enforced physical distance, etc.), curfews, quarantine, and home isolation measures, mandatory use of face masks, accompanied by massive deployment of disinfectants, supply of contact tracking mobile-device applications, and diagnostic tests [16–18]. Most commonly used are PCR tests that detect the virus in nasopharyngeal swabs, diluted gargle samples, or peripheral blood. As PCR tests amplify virus-specific RNA, they are characterized by very high specificity. The sensitivity of such tests varies across different products on the market. Moreover, PCR tests are easy to perform. Alternatives to quantitative PCR tests are CRISPR-based [19, 20], which are rapidly performed, and have high specificity and sensitivity. Other tests are antibody or antigen based. Such tests are less specific and do not necessarily detect active infections, since antibodies and antigens are present in the blood serum after the infection is cured.

The WHO established guidelines—including regular testing of employees and residents—to protect individuals in LTCFs [21, 22] due to high case fatality rates [23]. In fact, every second COVID-19 related death occurred within LTCFs in Belgium, France, Germany and Sweden [24, 25]. The situation was similar in other countries in the European Union, in which 37-66% of all COVID-19 related deaths occurred inside LTCFs [25]. The highest percentage of

66% was reported in Spain [25]. In absolute numbers, by May 2020, e.g., 14 000 residents and over 40 000 LTCF staff have been infected with the SARS-CoV-2 virus in France, while 8 935 staff have been infected [25]. In the U.K. and Scotland, respectively, 6% and 47% of all COVID-19 related deaths occurred inside LTCFs [25]. In the U.S. 39% of COVID-19 related deaths occurred among LTCF residents and staff [26]. Here, LTCFs refer to facilities/institutions, in which residents with difficulties living independently stay for long periods. They are cared for by staff members who do not necessarily reside in these facilities but live outside and interact with the general public. They subsume facilities such as retirement homes, skilled nursing facilities, assisted living facilities and residential care facilities [25]. A large number of these residents suffer from chronic medical conditions related to aging or have different impairments. Moreover, such facilities also subsume incarceration facilities (IF). Namely, residents are also confined to the facility, share risk factors, and they interact with staff (e.g., prison guards) that live outside the facility and interact freely with the outside world. However, the contact behavior in IFs will be very different from that in, e.g., nursing homes (see also Discussion). In the following we use the terms LTCF and IF to refer to stereotypical nursing homes or prisons. The reader should keep in mind that both terms can be used interchangeably when referring to the institutional structure, but not when referring to the contact behavior.

Residents of LTCFs constitute a substantial group in high-income countries such as Germany. With a population of 82.79 million, the number of people depending on nursing either in LTCFs or at home in Germany increased from 2.5 million in 2011 to 3.41 million in 2017 (over 66% of them being over 90 years old) [27]. The capacity of LTCFs in Germany was 952 367 beds (full stationary capacity: 885 488) in 2017, with 743 120 beds (723 451 full stationary) filled (623 182 beds in 2011, 612 183 being full stationary) [27]. These are sustained by 764 648 employees, 64% of which are care and support personnel [27]. These numbers have an increasing trend: there were additional ambulant care services supported by 829 958 people in need of nursing with 390 322 employees [27].

Residents in IFs also do not interact freely with the general community and are overseen by staff members, who do not permanently reside in these facilities. IFs face challenges in controlling the spread of COVID-19 [28, 29], and put elderly at particular risk of severe infection [30]. Indeed, jurisdictions in the U.S. have accelerated the release of low-risk offenders [31] as a measure to mitigate COVID-19. An estimated number of 6.4 million individuals were held under the supervision of the U.S. adult correctional system in 2018 (including probation and parole), with an incarcerated population of approximately 1.4 million [32–34]. There is a notable increase in the age structure of state prisoners, in which the numbers of inmates older than 55, 60, and 65 years of age have quadrupled from 1993 to 2003 [35]. Notably, COVID-19 rates in U.S. prisons are four times higher than in the general population with about 20% of inmates having being tested positive by December 2020 [36]. Residents in IFs share several risk factors with LTCFs such as underlying health problems, psychological isolation etc. E.g. in the U.S. chronic health conditions are particularly prevalent compared to the general population [37]. Apart from the similarities, the contact behavior and possibilities to implement hygienic measures and physical distancing are very different in LTCFs and IFs.

For both LTCFs and IFs the use of routine screening of residents, employees, and visitors by diagnostic tests before entering the facility to protect the resident risk groups was mentioned in guidelines by public health authorities [21, 22, 38, 39]. The effectiveness of such interventions is unclear but can be predicted by mathematical models.

Here, a mathematical model, based on the freely available CovidSIM simulation tool, is adapted to estimate the benefit of routine screening for COVID-19 infections of employees in LTCFs and IFs by regular tests. The same model applies to both LTCFs and IFs. However, parameters need to be chosen differently to adequately reflect the typical contact behavior in

both institutional types. Here, we focus on studying the impact of (i) the frequency at which employees are tested, (ii) the processing time to obtain test results, and (iii) the quality of the test in terms of sensitivity. While the model is described verbally in the main text, a concise mathematical description can be found in the S1 Appendix. The model is exemplified by parameters that reflect roughly the situations of LTCFs in Germany and IFs in the U.S. These, countries were chosen as examples because LTCFs and IFs are of particular importance in Germany and the U.S., respectively. The model per se is applicable to other industrial nations and other closed facilities.

## Materials and methods

We study the impact of testing employees in LTCFs or IFs to protect resident risk groups from COVID-19 infections using an extended SEIR model, i.e., by a deterministic compartmental model of ordinary differential equations. In particular, the model is an extension of that underlying the pandemic preparedness tool CovidSIM [40]. The flow chart of the model is presented in Fig 1. The model is described verbally with the concise mathematical description found in S1 Appendix. For the sake of simplicity, in the description, we focus on LTCFs. The model is equally applicable to IFs, because the structure of the model remains unchanged. Only the model parameters need to be tailored to the specific type of institution.

A population of size $N$ is divided into three groups: (i) the resident risk group (Ri), i.e., residents of LTCFs, (ii) the employees (staff) working in LTCFs (St), who are in close contact with the risk group, (iii) and the general population (Ge), i.e., the rest of the population.

Each group (Ge, St, Ri), is further subdivided into susceptible, infected, recovered, or dead individuals. Infected individuals pass through: (i) a latency period (not yet infective), (ii) a prodromal period (already partly infective but not yet exhibiting characteristic symptoms), (iii) a fully infectious period (symptoms ranging from non-existent or mild to severe), and (iv) a late infectious period (no longer fully infectious). All individuals either recover from COVID-19 and obtain full permanent immunity or die. The model follows the change of the number of individuals, per unit time, being in the susceptible ($S$), latent ($L$), prodromal ($P$), fully infectious ($I$), and late infectious ($L$) periods, and in the final recovered ($R$) and dead ($D$) stages separately for the three population subgroups (Ge, St, Ri). Deaths unrelated to COVID-19 are ignored, as we assume a pandemic in a large population in a relatively short time period.

In classical SEIR models, individuals in the latent, prodromal, infected, and late infected classes simply proceed from one stage to the next at a rate directly related to the residence time in each stage. This simplistic flow implicitly assumes that the times individuals spend in the various compartments are exponentially-distributed, and hence have a large variance. In particular, a proportion of individuals progresses too fast, whereas others progress much too slow.

To resolve this, we subdivide the latent, prodromal, fully infectious, and late infectious periods into several sub-stages, through which individuals pass successively. This ultimately leads to more realistic durations and hence dynamics.

The characteristics of the population subgroups (Ge, St, Ri) are incorporated within the contact behavior. Namely, the risk group has mainly contacts with other individuals in the risk group and the LTCF employees, whereas their contacts with the general population are limited. The employees (St) have contacts among themselves, with the risk group and the general population. However, the general population has mainly contacts among themselves. Given a contact within or between certain sub-populations, the contact occurs at random. The contact behavior is captured by the contact matrix (see S1 Appendix section "The basic reproduction number and the next generation matrix").

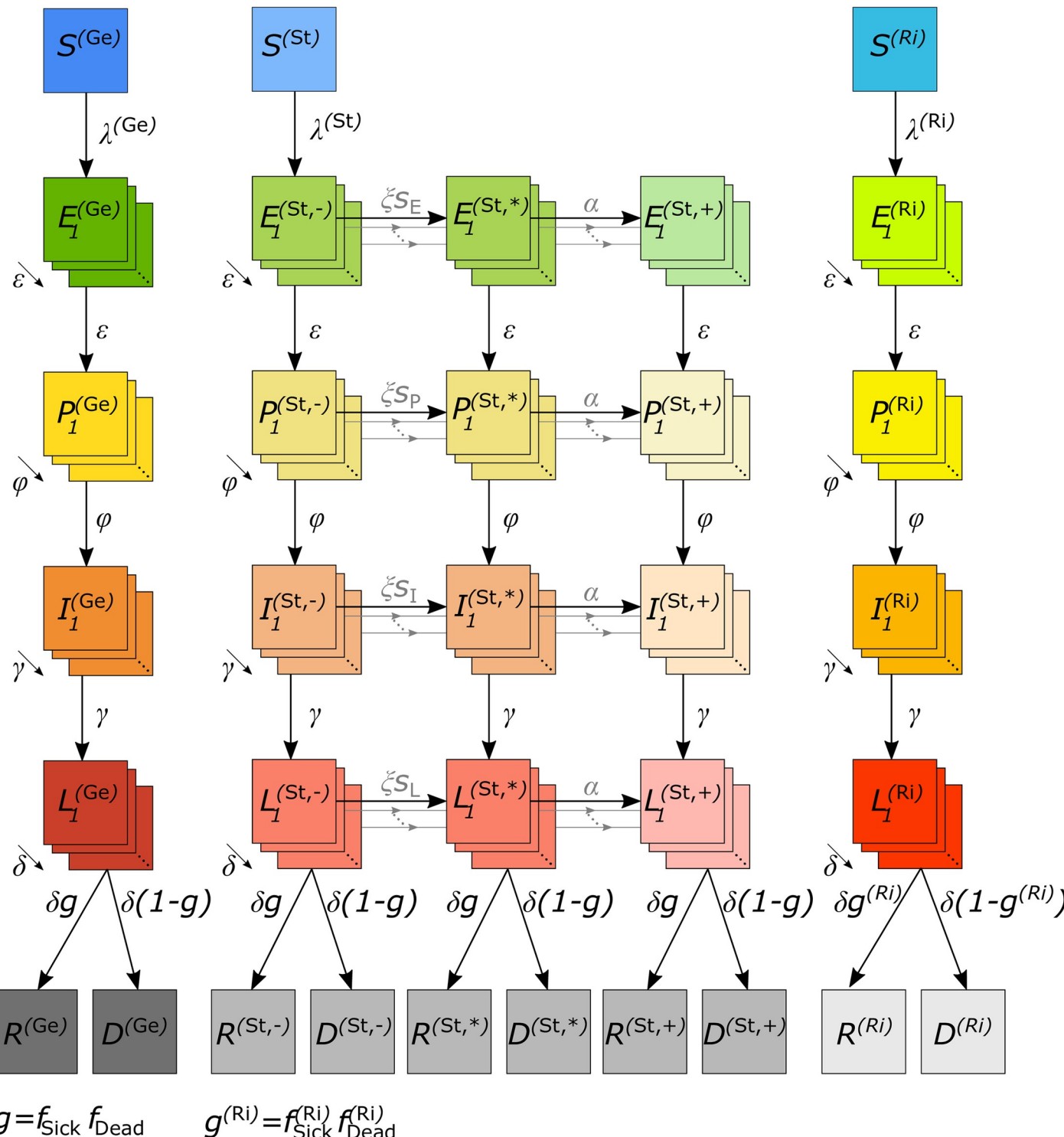

$$g = f_{\text{Sick}}\, f_{\text{Dead}} \qquad g^{(Ri)} = f_{\text{Sick}}^{(Ri)}\, f_{\text{Dead}}^{(Ri)}$$

**Fig 1. Model flow chart.** The total population is divided into three main groups, the general population (Ge), the employees of the long term care facilities (St), and the residents of those facilities (Ri). Infection flows between members of each group as explained in the text.

Susceptible individuals acquire infections through contacts with individuals in the prodromal, the fully infectious, or the late infectious periods at rates $\beta_P, \beta_I, \beta_L$, respectively, which are identical for each subgroup.

The basic reproduction number $R_0$ is the average number of infections caused by an average infected individual in a completely susceptible population during the infectious period. In a subdivided population (here Ge, St, Ri), the definition of $R_0$ is not straightforward and has to be derived from the next-generation matrix [41] (see S1 Appendix section "The basic reproduction number and the next generation matrix"). Importantly, $R_0$ fluctuates seasonally with a yearly average basic reproduction number of $\bar{R}_0$. Seasonal fluctuations in $R_0$ are motivated by fluctuations in the intensity of UV radiation, temperature, and time spent inside closed rooms [42], and intuitively supported by the spike of COVID-19 infections in European counties in fall/winter 2020.

First, infected individuals are latent carriers, before they enter the prodromal period, in which they become partly infective. From there, they enter the fully infectious period, at the beginning of which, it is determined whether the infection proceeds as symptomatic or asymptomatic. The likelihood to suffer from a symptomatic episode is elevated in the risk group (Ri).

Covid-19 confirmed individuals are subject to case isolation. Specifically, a fraction of symptomatic individuals will be detected and isolated in quarantine wards (perfect isolation preventing all contacts). If the wards are full, infected individuals are sent into home isolation (imperfect isolation, preventing only a fraction of contacts). Regarding this, there are differences in the subgroups: each symptomatic individual in the risk group will be detected and isolated in quarantine inside the LTCF (perfect isolation), whereas only a fraction of the general population and the LTCF employees will be isolated. Infected individuals further progress to the late infectious stage, during which they will stay quarantined as determined during the fully infectious stage. Importantly, LTCF employees will be tested for COVID-19 on a regular basis. We assume that the test is 100% specific, i.e., there are no false-positive test results, reflecting PCR- or CRISPR-based tests [19, 20]. If tested positive, they will be isolated either in quarantine wards or at home, in which case all contacts with the risk group are prevented. Staff can be tested positive during any of the infected stages (latent, prodromal, fully infectious, late infectious), however with different sensitivity depending on the characteristic of the COVID-19 test being used, irrespective of symptoms. In particular, there is a possibility (depending on the sensitivity of the COVID-19 test) that employees are isolated already during the latent period before they are infectious. Test results are not obtained instantaneously, but with a time delay. Infected staff can still infect others during this time. The waiting time for the test results depends on the available infrastructure.

Finally, late infected individuals, either recover or die. Only symptomatic infections can cause death. The fraction of lethal infections is higher in the risk group.

All case isolation mechanisms are not initially present, but implemented with a time delay after the initial occurrence of the disease. In addition to case isolation, general contact reduction interventions, e.g., social distancing, curfews, etc. will be sustained in a time-dependent fashion, reflecting lock-downs and other contact-reducing interventions. Importantly, contact reduction does not affect all individuals in the same way. Contacts within and between the population sub-groups are affected differently, reflecting the specific kind of reductions being implemented. During the time interval in which case isolation measures are sustained, contacts between the risk group and the general population are reduced, reflecting preventative measure. Furthermore, visitors have to provide a negative test result before entering the LTCF. To obtain conservative estimations the latter intervention is ignored in the simulations.

## Implementation of the model

The model following the description in S1 Appendix was implemented in Python 3.8 with a 4th order Runge-Kutta method using the function solve_ivp as part of the library Scipy (the Python code is available at https://github.com/Maths-against-Malaria/COVID19_LTCFs.git and http://doi.org/10.5281/zenodo.4634314). Graphical output was created in R [43].

## Results

The effect of protecting a resident risk group (LTCF residents) by regularly testing LTCF employees, who are the most likely to import the disease into the facility, is studied. The focus is mainly on PCR tests, but highly sensitive antigen tests are also assumed. Model parameters are adjusted to roughly reflect the situation in the Federal Republic of Germany, one of the countries that has successfully intervened in the COVID-19 epidemic. The model itself is applicable to any industrial nation with an aging population. The aim is to investigate the effects of protecting the risk group and to estimate the demand for (PCR) tests. Some testing scenarios are not feasible in terms of logistics and available testing capacities and just serve as a comparison.

We also apply the model to IFs, by parameterizing it to reflect roughly the situation of U.S. prisons. We treat the case of LTCFs first, and present the case of IFs as a comparison.

The parameters used for Germany are listed in Table 1, and S1–S6 Tables. Germany has a population of roughly $N = 83$ million. We assume, 700 000 elderly individuals live in LTCFs in which control interventions by PCR testing can be implemented. All employees of LTCFs amount to approximately 760 000. This number however includes employees in the administration and external services, who are not regularly working in these facilities. Hence, a number of 500 000 employees was assumed to work regularly in the LTCFs. The first COVID-19 cases were introduced in the middle of February 2020, corresponding to time $t = 0$. A seasonal average basic reproduction number of $\bar{R}_0 = 3.4$ was assumed, which fluctuated seasonally by 43% over the year, with a peak roughly in late December ($t_{R_{0_{\max}}} = 300$). In Germany estimates for $R_0$ varied substantially, with 4.43 (95% CI 1.83–7.92) in March 2020 [44]. Here $R_0$ was chosen to yield realistic dynamics under the contact reductions imposed in Germany. Additionally, the average duration of the latent, prodromal, fully infectious and late infectious stages were assumed to last on average $D_E = 3.7$, $D_P = 1$, $D_I = 5$, $D_L = 5$ days, respectively. In the prodromal and late infectious stages, individuals were assumed to be half as infective as in the fully infectious stage. Individuals in the risk group were more likely to develop severe symptoms ($f_{\text{Sick}} = 58\%$ *vs.* $f_{\text{Sick}}^{(\text{Ri})} = 60\%$) and had an increased mortality ($f_{\text{Dead}} = 1.6\%$ *vs.* $f_{\text{Dead}}^{(\text{Ri})} = 20\%$). The effect of restricting LTCF access to members of the general population that provide a negative COVID-19 test result upon entry is ignored in the simulations.

**Table 1. Test sensitivity.** Sensitivity of tests with different quality in the respective stages of the infection.

| test | $s_E$ | $s_P$ | $s_I$ | $s_L$ |
|------|-------|-------|-------|-------|
| poor PCR | 0.00 | 0.10 | 0.65 | 0.35 |
| intermediate PCR | 0.03 | 0.30 | 0.75 | 0.50 |
| good PCR | 0.15 | 0.60 | 0.80 | 0.60 |
| very good PCR | 0.25 | 0.75 | 0.90 | 0.65 |
| excellent PCR | 0.30 | 0.80 | 0.95 | 0.85 |
| antigen | 0.00 | 0.35 | 0.85 | 0.85 |

General contact reduction was chosen intuitively to reflect approximately the interventions imposed in Germany over time. The reductions were chosen such that the overall observed disease prevalence (accounting for undetected infections) was accurately reflected until February 2021. An additional "hard lockdown" was assumed to be extended until April 2021 followed by a "soft lockdown" until summer 2021 ($t = 450$). After that we assumed no contact reductions within the general population, and the general population and the LTCF staff for three reasons: (i) with the ongoing vaccination campaigns, compliance with contact reductions might be low after summer 2021; (ii) this allows us to study the impact of the intervention during different waves (and is informative about what could have been achieved in the past and what can be achieved in the future); and (iii) we want to illustrate what happens in a well-managed epidemic *vs.* an ill-managed epidemic without strong contact reductions. The latter reflects a strategy aiming for herd immunity, i.e., no interventions in the majority of the population, while protecting only risk groups. For comparison we also assumed a more radical herd-immunity scenario with no general contact reductions in the whole population after $t = 450$ and protecting the risk group just by regular testing of staff. In any of the scenarios the epidemic has three waves with three peaks: (i) the initial wave in April 2020, (ii) the second wave that started in late September 2020 and peaked in December 2020, and (iii) a hypothetical third massive peak in late October 2021 corresponding to a herd-immunity strategy—this scenario is intentionally unrealistic.

Clearly, the available capacities of tests, the infrastructure to perform tests, the waiting time for results, and the sensitivity of tests can vary substantially. The impact of these factors is investigated. In any case, testing hardly affects the general population, while it has an effect which can be profound on the risk group or even the staff (see Figs 2–8). Typically, low sensitivity of tests can be compensated by testing more frequently to reduce the number of infections in the risk group as illustrated below by comparing PCR with antigen tests (see also Figs 8 and 9). The effects of various aspects of the testing intervention are explained below. Importantly, testing interventions are assumed from the onset of the epidemic to study the hypothetical effect that could have been achieved by screening LTCF staff.

## Testing rate

Assuming a baseline "good quality" PCR test with a processing time of 48 hours, the intervention has a profound effect on the number of infections and deaths in the risk group as long as the number of infections is not too high (see Figs 2 and 3), reducing the number of infections at the epidemic peak in the second wave by about 35% (see Fig 2E) and the number of deaths by a similar percentage (see Fig 3E). Testing daily would lead up to a 55% reduction. This is not true for the first wave, because initial infections were mainly introduced from outside the LTCF before general contact reduction and case isolation were implemented. During the first wave testing is much less effective. During the third peak (Figs 2F and 3F), even daily testing leads only to a relative insignificant reduction of infections. However, the reduction in mortality is still noticeable and about 11% for testing every 5 days. Testing is slightly more efficient in relative (but not in absolute) terms if no general contact reduction is assumed at all (Figs 2H and 3H).

The results for the staff are similar to those for the risk group in the first and second wave, however, the reduction in the number of infections (Fig 2C and 2D) and (particularly) deaths (Fig 3C and 3D) are less pronounced. The epidemic peak in the second wave is reduced by 30% when testing every 5 days (cf. Fig 2C with 2E). Testing daily would lead up to a 45% (cf. Fig 2C with 2E). Importantly the number of infections is higher in the third wave—the higher the testing rate, the higher the increase in the number of infections (see Fig 2D). The reason is

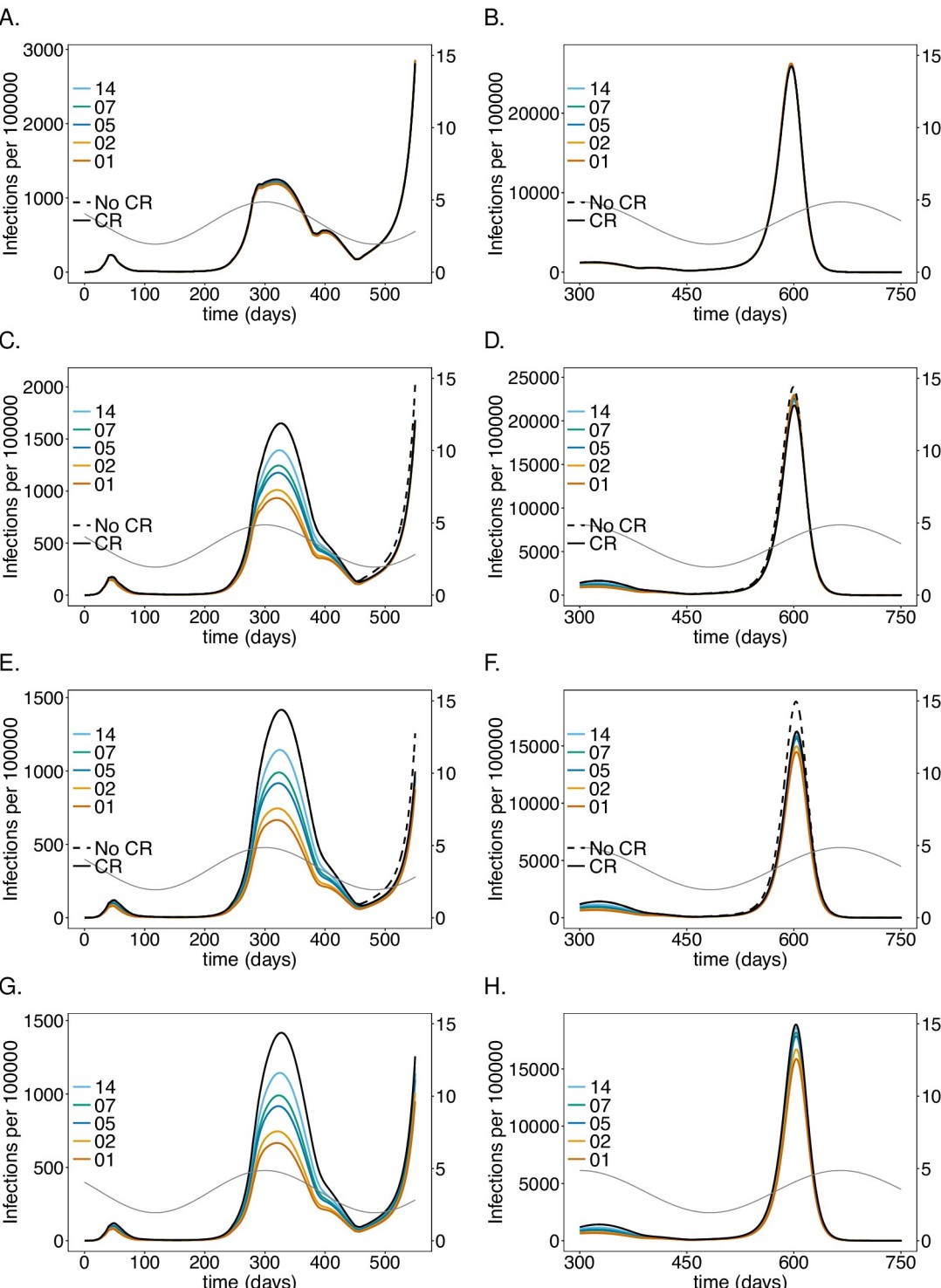

**Fig 2.** Impact of rate of testing LTCF staff on the number of infections: Shown are the numbers of infected (*I*) individuals per 100 000 individuals (normalized to the sub-population size) at time *t* for (A-B) the general sub-population (Ge), (C-D) the LTCF staff (St), and (E-F) the risk group (Ri), assuming different testing rates. A good-quality test (cf. Table 1) is assumed with a processing time of 48 hours. After generation *t* = 450 no contact reductions among Ge, and between Ge and St are assumed. Only contact reductions within LTCFs are assumed. The solid black lines correspond to the baseline model without the testing intervention. The dashed black lines show the number of infections if neither testing nor any general contact reductions is assumed after generation *t* = 450. Panels G, H show the number of infections for corresponding simulations assuming no

contact reduction in the LTCFs after time $t = 450$—the solid black line corresponds to no testing. Panels A, C, E, G show the dynamics for the time range from $t = 0$ to $t = 550$, while the remaining panels show the dynamics for the time range from $t = 300$ to $t = 750$ (note the difference in the range of the y-axes for the different time ranges). Seasonal fluctuations in $R_0$ are assumed and are indicated by the dashed grey lines corresponding to the axis on the right-hand side in all panels. The parameters used for the simulations are listed in Table 1, and S1–S6 Tables.

as follows. Initially less individuals are infected, leading to a larger susceptible population, such that the momentum of the third peak as $R_0$ increases due to seasonal fluctuations is higher, leading to a stronger exponential growth in the number of infections. In fact, the third peak is higher than in the case without testing. However, mortality is still slightly reduced (Fig 3D). For daily testing the third peak is almost as high, as if neither tests were performed, nor contact reductions sustained after time $t = 450$.

The general population is hardly affected by testing the LTCF staff. The number of infections are slightly lower in the second wave, for which the intervention is most successful. The third peak increases with increasing testing rate for the same reasons as for the LTCF staff (Fig 2A and 2B). Overall mortality is not noticeably affected (Fig 3A and 3B).

Notably, the number of infections per 100 000 is largest in the general population and smallest in the risk group. Cases, however, are easier to detect in the risk group, because the fraction of symptomatic cases is higher. Mortality per 100 000 individuals is much higher in the risk group than in the general population and among LTCF staff.

## Test-processing time

The time needed to process PCR tests, reflecting the testing infrastructure, has a similar effect as the testing rate (see Subsection Testing rate) during the first two waves, assuming weekly testing with a good-quality PCR test every 5 days (see Figs 4 and 5). When comparing test-processing times of 0.5-4 days (12, 24, 48, 72, 96 hours), shorter ones lead to a noticeable reduction in the number of infections, particularly there is a noticeable gain when shortening the processing time from 48 to 24 hours. Unlike for the testing rate, the test processing time leads to a reduction in the third peak for the following reason: the momentum of third wave is not captured completely because the number of susceptibles was sufficiently reduced as the processing time is not as relevant in reducing incidence as the testing rate.

The effect of the processing time is similar in the LTCF staff population, but less pronounced as in the risk group. The general population is again hardly affected.

## Test sensitivity

The sensitivity of the tests, which varies between the stages of the infection, impacts the number of infections and deaths in the risk group and staff (Figs 6 and 7). The effects are similar to those of the testing rate and processing time (see Subsections Testing rate, Test-processing time). Testing every 5 days with a test processing time of 48 hours was assumed. For simplicity, we compared only tests, which had higher or lower sensitivity across all stages of the infection as specified in Table 1. During the second wave, improving the quality of the test from poor (failure to detect the virus during the latent phase, with a maximum sensitivity of 65%) to intermediate (maximum sensitivity of 75%) yields a 30% instead of a 25% reduction in the number of infections in the risk group. A good-quality test (maximum sensitivity 80%), yields a 35% improvement. The effect of increasing sensitivity eventually saturates. The relative reduction of infections in the third peak is not pronounced.

Similar results hold for the staff, however, the number of infections increases with test quality during the third wave, for the same reason as higher testing rates lead to a higher peak. The

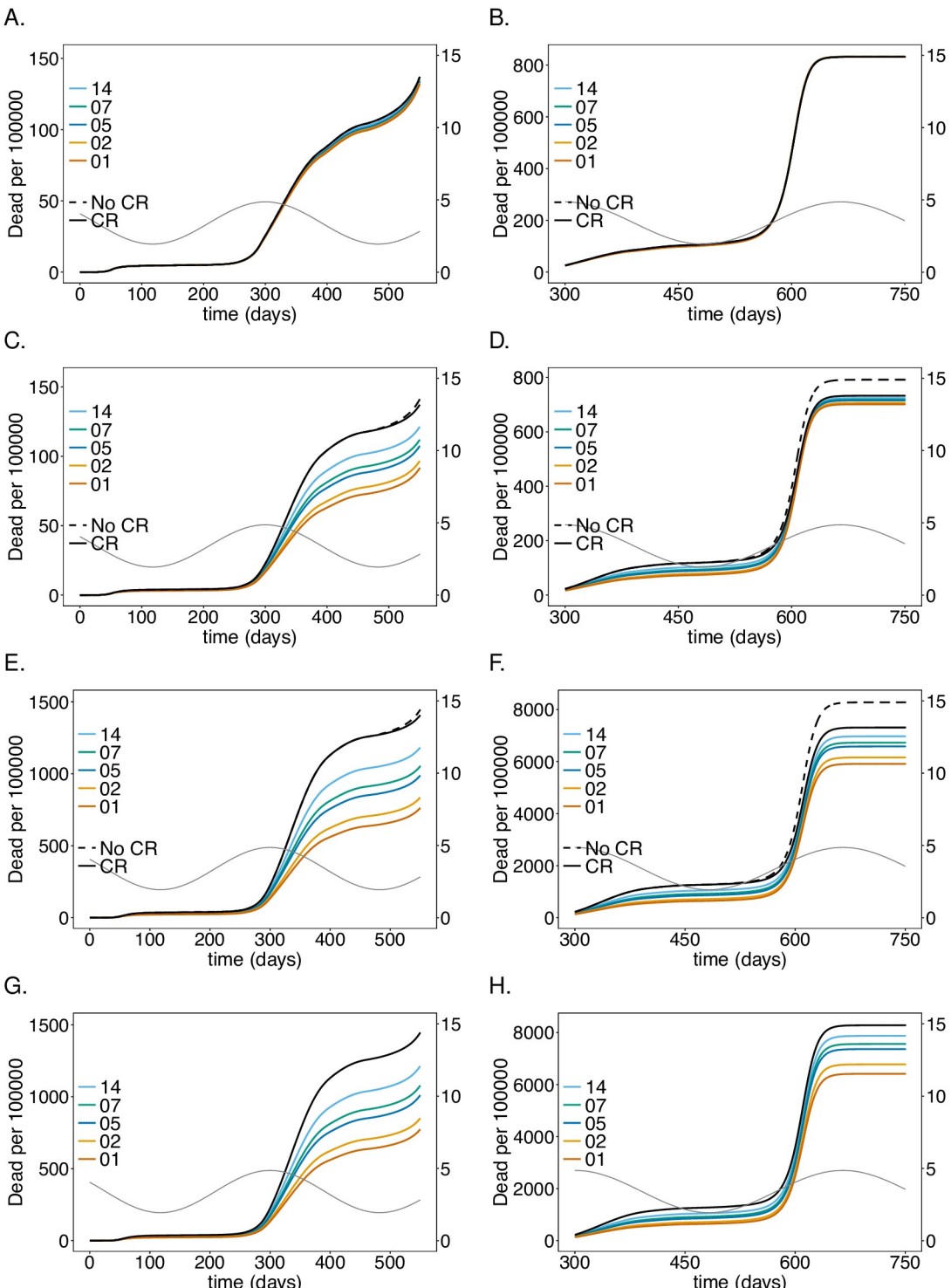

**Fig 3. Impact of rate of testing LTCF staff on mortality: As in Fig 2 but for (cumulative) numbers of deaths (*D*) at time *t*.**

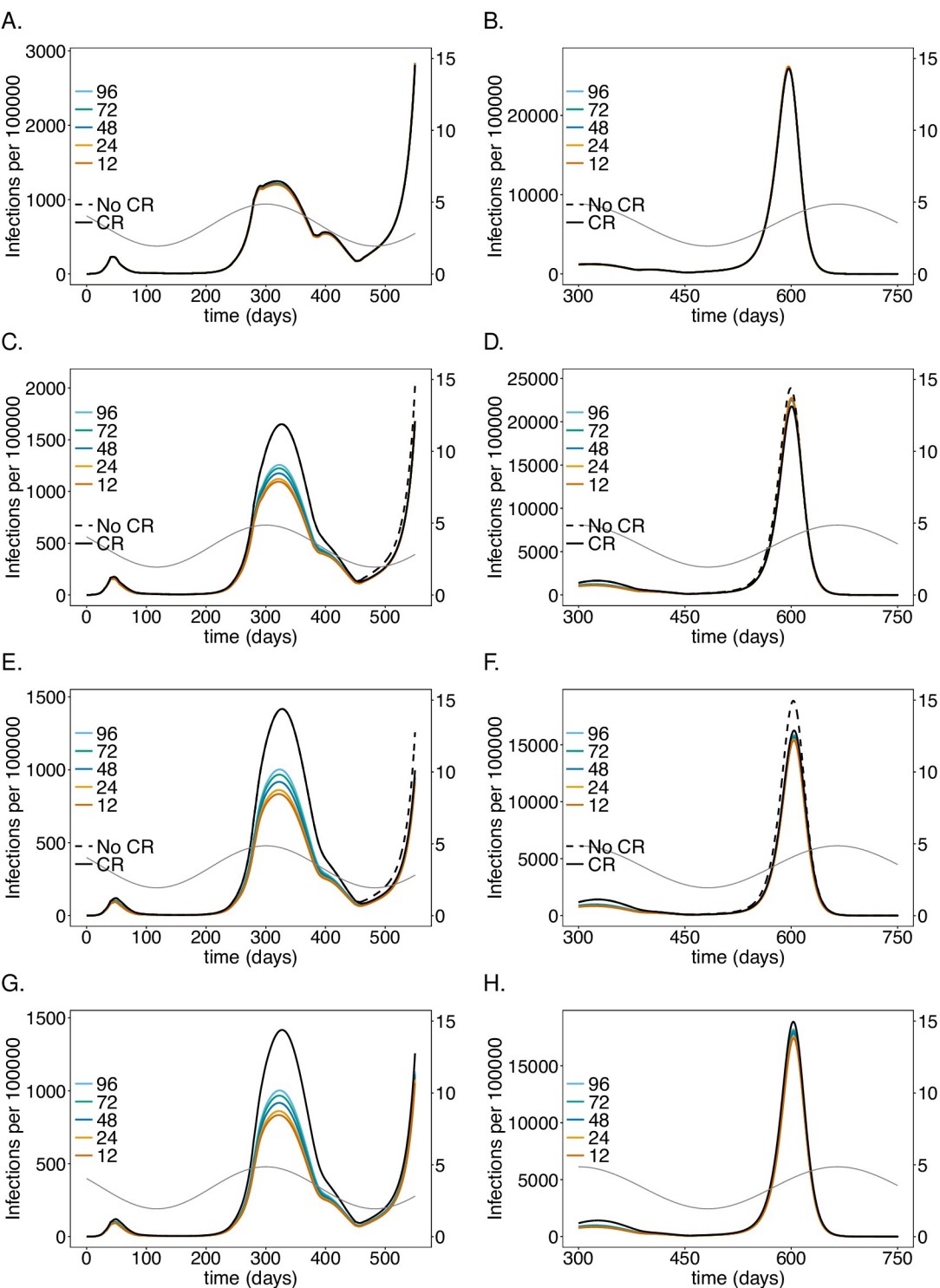

**Fig 4. Impact of processing time for testing LTCF staff on the number of infections: As in Fig 2 but for different test processing times assuming a testing rate of once per 5 days with a good-quality test (cf. Table 1).**

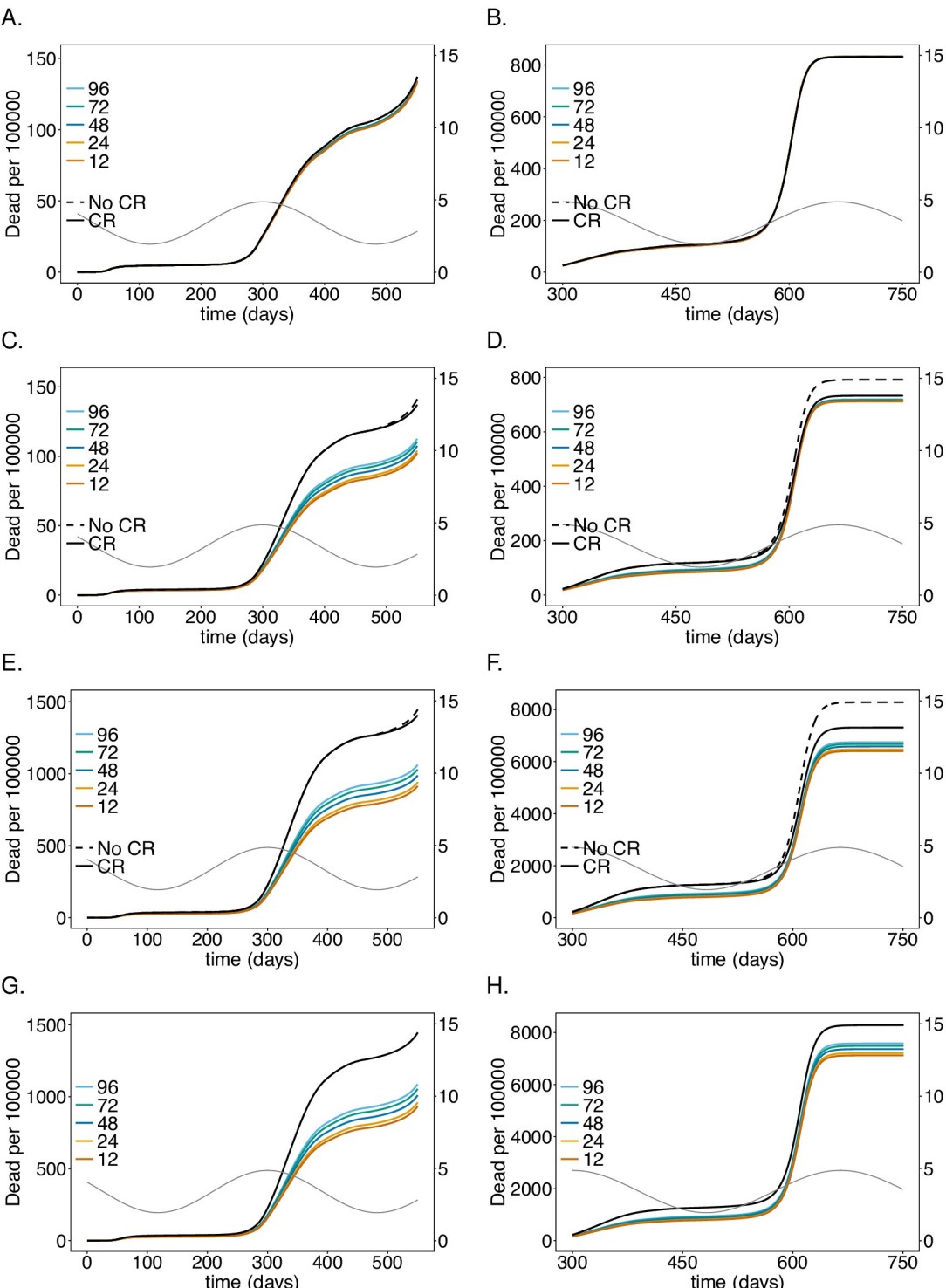

**Fig 5. Impact of processing time for testing LTCF staff on mortality: As in Fig 4 but for (cumulative) numbers of deaths (D) at time t.**

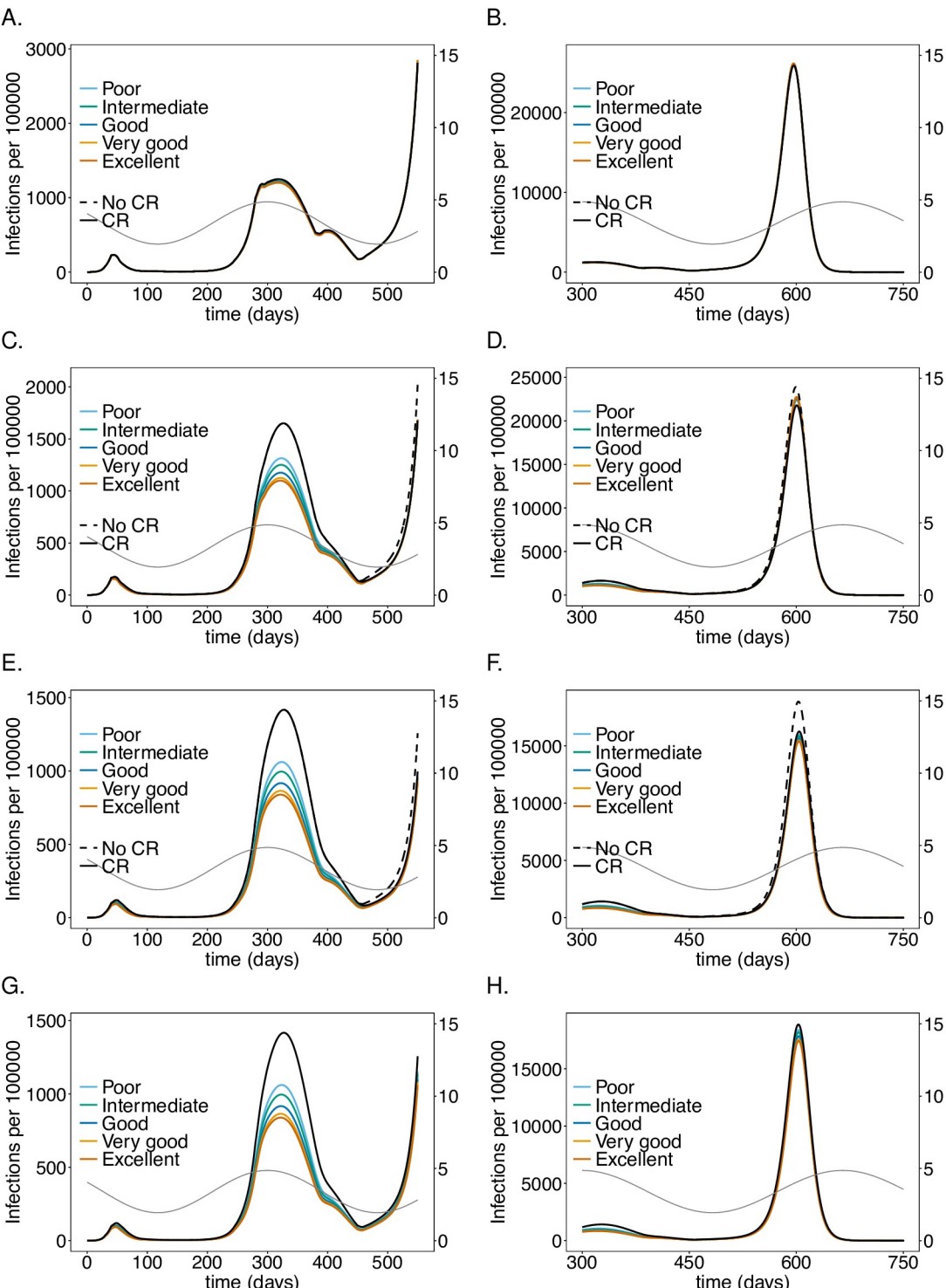

**Fig 6. Impact of sensitivity when testing LTCF staff on the number of infections: As in Fig 2 but for different test qualities (determined by their sensitivity; cf. Table 1).**

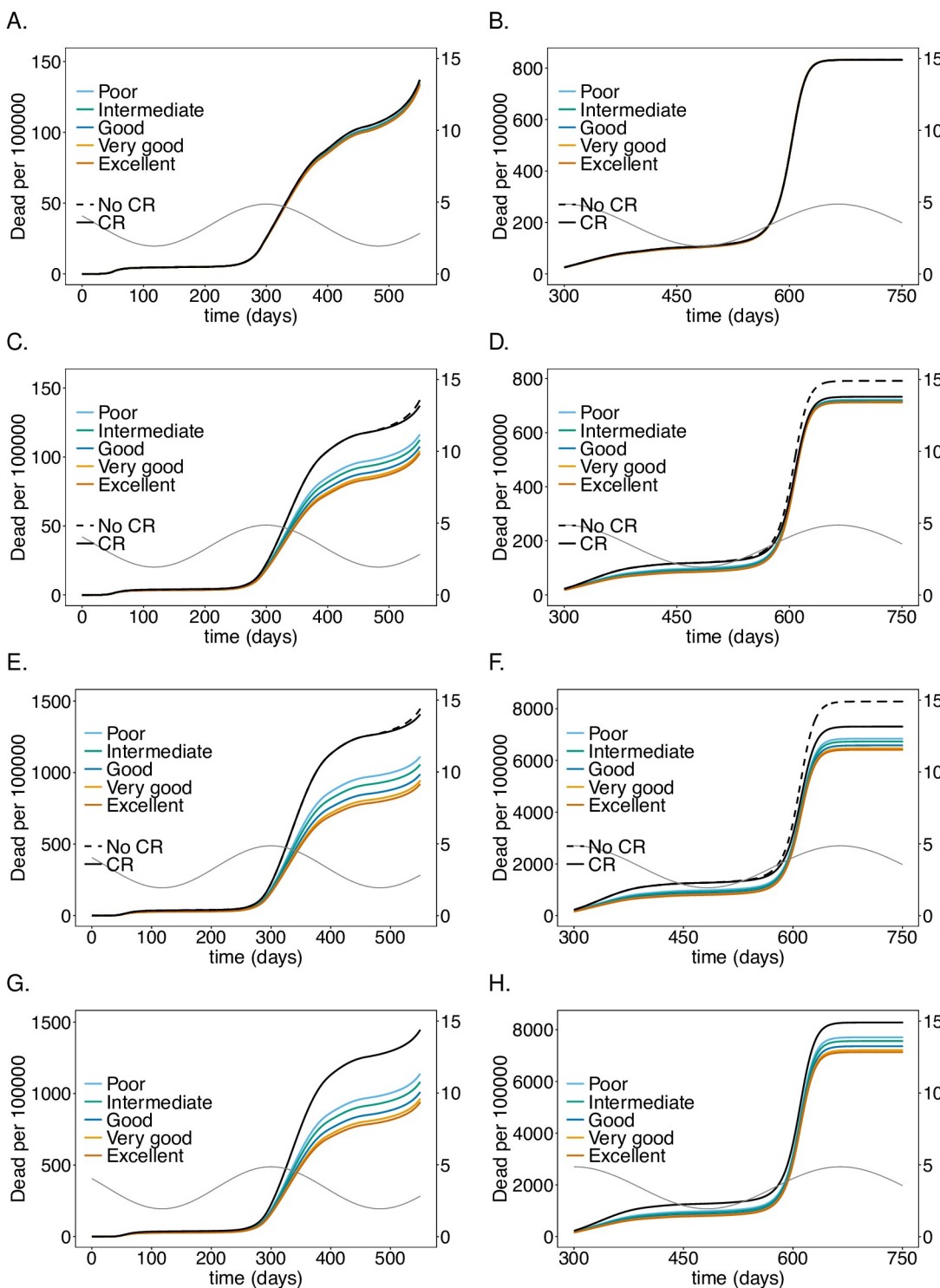

**Fig 7. Impact of sensitivity when testing LTCF staff on mortality: As in Fig 6 but for (cumulative) numbers of deaths (D) at time t.**

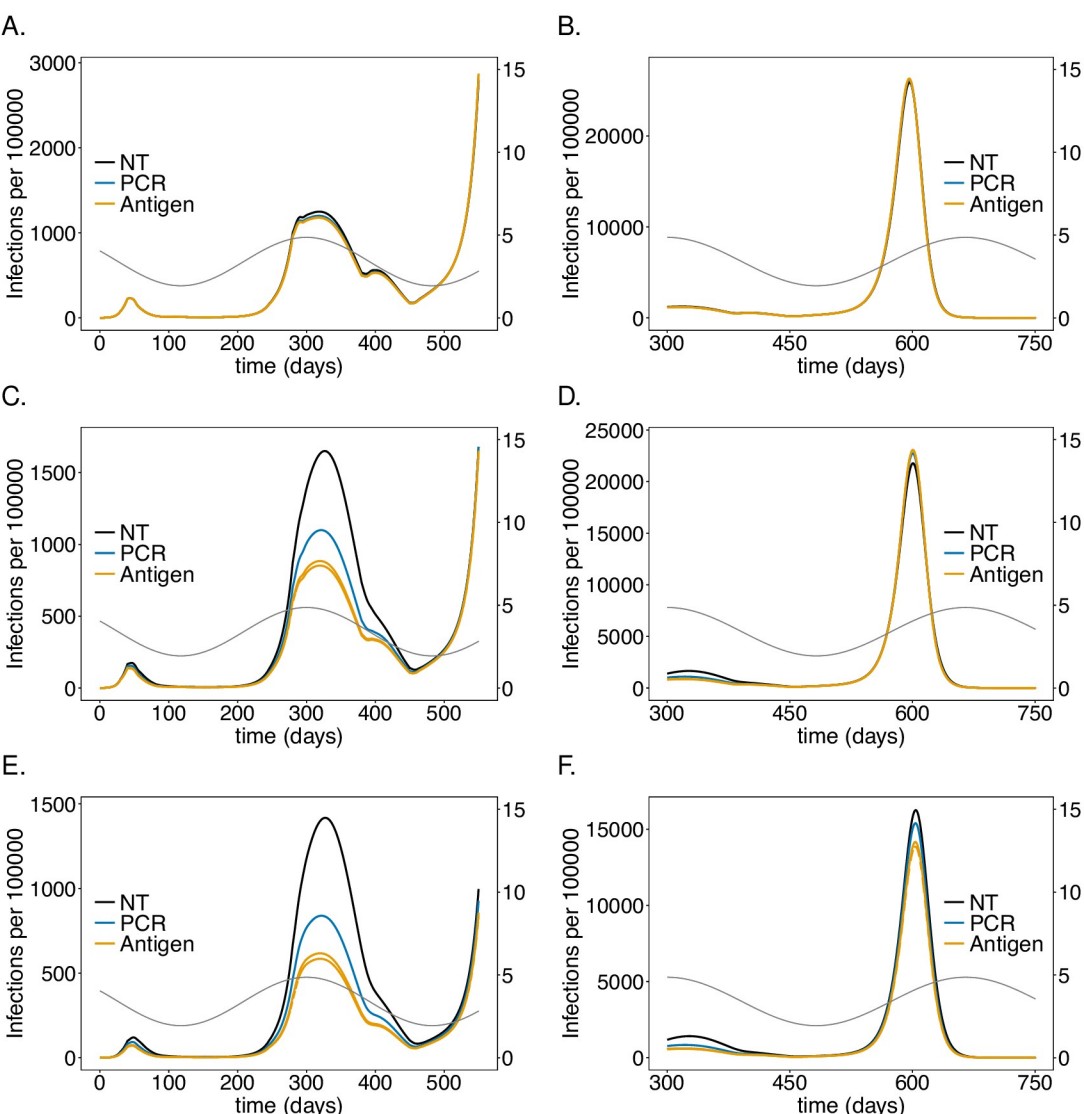

**Fig 8. Impact of PCR *vs*. antigen tests on the number of infections: As in Fig 2 but comparing testing using PCR and antigen tests with no testing (colors).** For the PCR scenario testing every 5 days with an excellent-quality PCR test (cf. Table 1) with a processing time is assumed. Antigen tests (cf. Table 1) are assumed to be performed one a day (solid) or twice a day (dashed) with a processing time of 15 minutes.

general population is hardly impacted—a slight effect is mainly noticeable during the second wave (Fig 6A).

## Antigen tests

In the model false positive tests are ignored, reflecting high specificity. While originally only PCR tests had such quality, antigen tests have been substantially improved [45–50]. While PCR tests need to be processed in a lab, their antigen-based counterparts are available as rapid tests that return the test result after 5-20 minutes. Moreover, it is feasible to perform them more frequently. However, antigen tests are less sensitive particularly in the early stages of the infection (but might be more sensitive in the late stages). As a comparison we investigated the impact of an antigen rapid test, specified in Table 1, that is performed either every day or twice

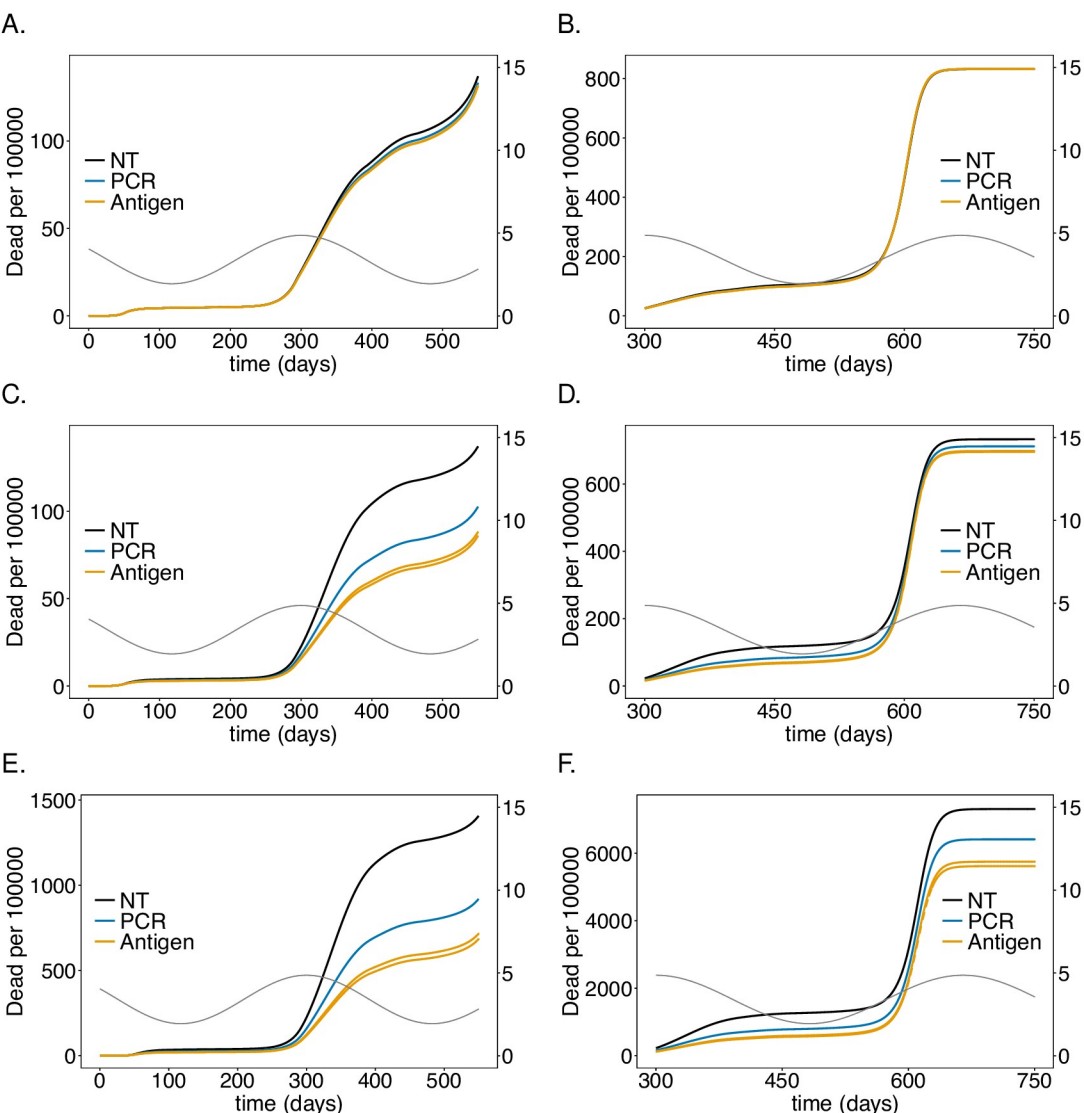

**Fig 9. Impact of PCR *vs*. antigen tests on mortality: As in Fig 8 but for mortality instead of the number of infections.**

per day and returns the result within 15 minutes with an excellent-quality PCR test that is performed every 5 days and returns the result within 48 hours. The results are shown in Figs 8 and 9.

The antigen test leads to a substantially stronger reduction in the number of infections and mortality in the risk group and among the LTCF staff. The effect in the general population is negligible. During the second wave antigen tests reduce the height of the peak by about 55% whereas PCR testing leads only to a reduction of approximately 40% (Fig 8E). The reduction during the third wave is smaller in relative terms, but the improvement compared with PCR-testing is still substantial (Fig 8F). The improvement of testing twice per day with antigen tests compared with daily testing is not pronounced. Overall the effect of reducing mortality during the second and third wave in the risk group and among LTCF staff is substantial (Fig 9C–9F).

The epidemic peak in the third wave is slightly higher in the staff population if antigen rather than PCR tests are used (Fig 8D).

## Economic considerations

Moral obligations set aside, the economic gain of the testing interventions can be derived from the results described above. Such considerations must incorporate testing capacities and the available infrastructure. From the simulations one can make an economic assessment of the costs of the intervention either from the onset of the pandemic or prospectively.

Treatment costs for an LTCF resident in Germany for respiratory diseases such as COVID-19 and other flu-like symptoms (compared in [51]) can be roughly estimated to range between 4 000-8 000 Euro from the official statistical reports on national healthcare costs [52]. Namely, as the influenza and pneumonia costs from men and women over 65 years of age in Germany in 2015 (1.41 billion Euro; cf. [52], section 4.3.5) divided by the number of full stationary influenza and pneumonia cases in hospitals and rehabilitation centers in the same age cohort (225 403; cf. [52] section 4.1.1 and 4.1.3). These estimates are in accordance with those for inpatient flu costs reported in [53]. Notably, the costs can only be a rough estimate, that does not include healthcare costs for resulting chronic respiratory conditions (which are a substantial fraction of the total healthcare costs).

According to the simulations roughly 256 000 LTCF residents will become symptomatically infected with the virus without the testing intervention (cf. Fig 8F). This number reduces to roughly 224 300 with PCR testing every 5 days. This results in savings of treatment costs of 126-251 million Euro. Assuming costs of approximately 40 Euro per PCR test, and testing once per working week (5 days; 48 weeks per year) over the whole duration amounts to 1.96 billion Euro. This assumes regular testing from the beginning of the pandemic. Hence, the costs for testing are an order of magnitude higher than the gain. Testing every working day (5 times per week; 48 weeks per year) with antigen test at a cost of 5 Euro per piece over the whole duration of the simulation is considerably cheaper at a cost of 1.24 billion Euro. However, the savings due to the reduction in the number of severe infections are considerably higher at a cost of 219-437 million Euro.

These calculations assume testing from the onset of the pandemic. One can also infer the future savings in healthcare costs. Between generation ($t$ = 350) and the end of the simulation ($t$ = 750), the PCR testing strategy would save 87-173 million Euro as opposed to 1.04 billion Euros in costs for tests. The costs for antigen tests would amount to 650 million Euro, while the savings add up to 162-325 million Euro.

These numbers are less discouraging than they appear at first sight. Namely, testing needs to be performed only in phases with high incidence, reducing the costs for tests substantially. Furthermore, the third wave is a hypothetical scenario that mocks a herd-immunity scenario that is unlikely to be politically sustainable. In practice, general contact reduction will reduce incidence and render testing more efficient. In addition, test quality is steadily improving. In any case, the considerations here serve just as an idea. The model should be fine tuned to specific situations of interest and more refined costs should be considered. Importantly, the considerations here focused on quantifiable aspects and did not include any moral obligations or long-term costs of infections (some of which cannot even be quantified at present).

## Incarceration facilities

The predictive model also applies to incarceration facilities (IFs). While, the basic setup is the same, demographic numbers, risk factors, contact behavior, and—importantly—the possibilities to implement protective measures are very different. Here we illustrate, how the model can be hypothetically adapted to U.S. prisons. As for German LTCFs we chose parameters that roughly reflect the observed number of infections over time (accounting for undetected cases).

Accurate estimations, would require reliable estimates of the contact-behavior in IFs. This requires in-depth knowledge that is beyond the scope of this article.

Model parameters are listed in Table 1, and S1–S5 Tables, and S7 Table. The U.S. has a population of approximately $N = 331$ million, of which 1.4 million are incarcerated (risk group) and guarded by 423 000 correctional officers [32, 33, 54].

The simulations start on January 20, 2020 ($t = 0$). A slightly smaller seasonal average basic reproduction number than in Germany, namely $\bar{R}_0 = 3.2$, was assumed (because of the lower population density). We further choose smaller seasonal fluctuations (35%), due to the country's lower latitude. The peak in $R_0$ was again late December ($t_{R_{0_{max}}} = 335$). Parameters describing the disease were chosen as for Germany. However, the fraction of the risk group to develop symptomatic infections $f_{\text{Sick}}^{(\text{Ri})} = 43\%$ ($vs.$ $f_{\text{Sick}} = 58\%$) was supposed to be lower than in the general population, due to the demographics in the risk group. On the contrary, the fraction of lethal symptomatic infections was assumed to be higher than in the general population ($f_{\text{Dead}}^{(\text{Ri})} = 11\%$ $vs.$ $f_{\text{Dead}} = 4\%$). This reflects, on the one hand, the overall younger population in IFs, and, on the other hand, the high number of pre-existing conditions and risk factors in the IF population.

Likewise for LTCFs, the choice of general contact reduction was intuitive to resemble the observed number of infections. Contrary to LTCFs, contact reductions were assumed to be more efficient in the general population than in IFs, where contact-distancing measures are notoriously difficult to implement [55]. Moreover, there might be a lack of compliance with distancing measures due to the behavioral patterns in prison [56]. Overall, contact reductions were assumed to be weaker than in Germany, reflecting the governments' different approaches in managing the disease [57]. As for Germany, no contact reduction between the general population was assumed after summer 2021 ($t = 450$).

The epidemic in the U.S. follows four waves (Fig 10, and S1–S6 Figs): (i) the initial wave in April 2020; (ii) the second wave in August 2020; (iii) the third wave in the holiday season (starting with thanksgiving holiday); (iv) the hypothetical fourth wave in autumn 2021 after contact reductions have been lifted (except those sustained in the IFs).

**Similarities and differences between LTCFs and IFs.**   The general population is less affected by testing staff than it was for LTCFs in Germany for the following reasons: (i) the risk group and staff populations constitute a smaller fraction of the population; (ii) the epidemic was/is stronger in the U.S., so control interventions are less effective; (iii) the interactions of the risk group with the general population is more restricted. Differences between German LTCFs and U.S. IFs arise mainly from the differences in contact behavior. More explicitly, residents in LTCFs have less contacts than the remaining population and it is relatively easy to comply with distancing and hygienic measures in such facilities. On the contrary, inmates in IFs have abundant close contacts due to overcrowding. Consequently, it is not possible to implement distancing and hygienic measures.

Testing significantly impacts the dynamics of the risk group and IF staff. There are a number of similarities compared with Germany, but also substantial differences. As for the case of LTCFs in Germany, the first wave of the epidemic is not noticeably changed by testing, and a strong reduction in the number of infections occurs in the second wave, due to the same mechanisms as described for Germany. The third wave in the U.S. corresponds to the third wave in Germany. However, due to seasonal fluctuations in $R_0$, the exponential growth in the number of cases is much stronger during the third wave if the number of infections was lower beforehand (during the second wave). While this effect was relatively moderate in Germany LTCFs, it is substantial in U.S. IFs. Because testing leads to much higher number of infections

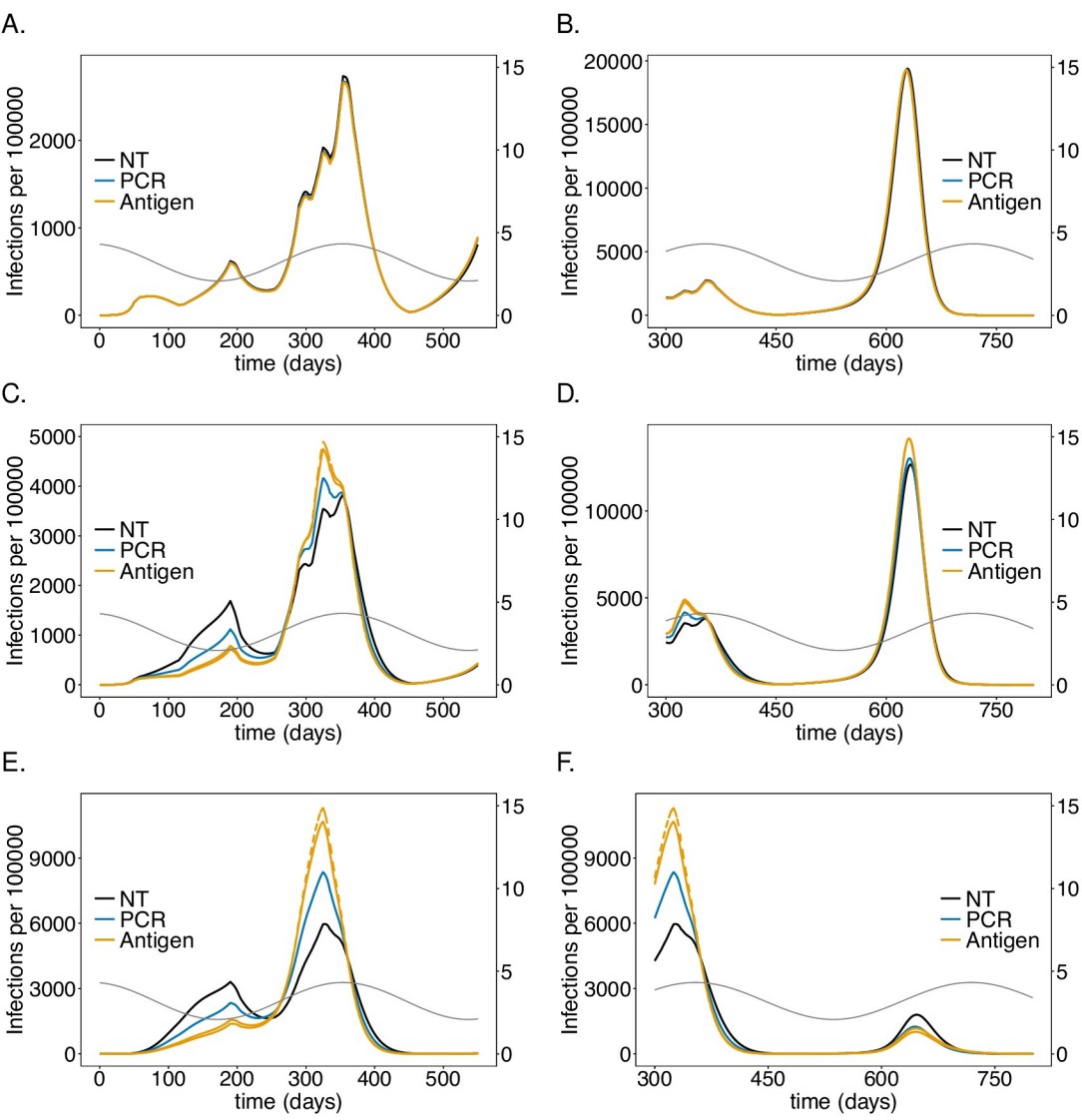

**Fig 10. Impact of PCR vs. antigen tests in IFs on the number of infections.** As in Fig 8 but for U.S. IFs instead of German LTCFs. Parameters for contact reduction are given in S7 Table.

during the third wave, the susceptible population is considerably reduced, such that the fourth wave is milder.

One difference compared with German LTCFs is that, the epidemic waves are less "synchronized" between the risk group, the staff and the general population. In fact, the number of infections within the staff and risk group are perfectly correlated, but different from the general population. Namely, the first wave is less pronounced, and blurs with the second wave, which is much more aggressive inside IFs. Indeed, the number of cases per 100 000 individuals is three to four times higher for the risk group and up to about two times higher for the IF staff (Fig 10, S1, S3 and S5 Figs). The same holds true for the third wave. The fourth wave, during which no contact reductions are assumed in the general population, is by far the most substantial one in the general population (Fig 10A and 10B). On the contrary, the third wave is much

milder for the IF staff, with an epidemic peak being about three times as high (Fig 10C and 10D). An important observation is made for the risk group: the fourth wave is much milder than the third wave and approximately as strong as the second one (Fig 10E and 10F). The reason is that the extraordinary high number of infections in the risk group during the third wave results in high levels of immunity.

**Testing rate in IFs.**  Regarding testing, the testing rate has the most pronounced effects. The effects of the testing rate in the first and seconds waves (S1 and S2 Figs) is similar as in German LTCFs. However, in the third wave the effect is much more pronounced, while the effect in the fourth wave is again similar to the one in Germany's third wave. The second and third waves in the U.S. correspond to 'controlled' epidemic peaks and is equivalent to the second wave in Germany. Importantly, testing in IFs leads to a much higher number of infections during the third wave due to the arguments outlined above. Hence, when considering the second and third wave as one compound, testing has undesirable effects for the risk group and staff due to the specific contact behavior in IFs. In particular, from the end of the third wave, testing leads to higher mortality (S2 Fig), which, however, changes in the short run. Importantly, in terms of mortality, the testing is much less effective than in LTCFs in the long run.

**Test-processing time in IFs.**  Similarly as in German LTCFs, the effect of processing time (S3 and S4 Figs) is less pronounced than that of the testing rate. The effects are similar: increasing the processing time has the same qualitative effect as increasing the testing rate.

**Test sensitivity in IFs.**  Also the test quality impacts the dynamics similarly as in German LTCFs. Improving the test quality (S5 and S6 Figs) has a stronger effect than increasing the processing time. As for German LTCFs this effect saturates.

**Antigen tests in IFs.**  Overall, testing hardly leads to a reduction in mortality, neither in the risk group nor among the IF staff. Testing more frequently with antigen tests, which has a beneficial and economically meaningful effect, in German LTCFs, leads even to a slight increase in mortality inside U.S. IFs (Fig 11F).

**Economic considerations in IFs.**  The simulations suggest as supported by empirical evidence [56] that the contact conditions in U.S. prisons render the risk group particularly vulnerable, and in fact expose them to high risk of infection. Interventions such as testing, which per se can have a beneficial effect (as for German LTCFs), can neither protect the risk group, nor compensate the lack of possibilities to implement appropriate precautionary measures. Economically, testing seems not to be cost-effective in U.S. IFs. The reason is that mortality and the number of infections is not strongly reduced in the long run. In the short run it causes even higher numbers of infections and mortality. Testing can only be part of epidemic management that promotes strong contact reductions, which cannot be realized in U.S. IFs due to overcrowding. Economic considerations must be part of rethinking infrastructure and conditions in IFs.

## Discussion

Elderly citizens and particularly residents of long-term care facilities (LTCF) were identified early as a vulnerable risk group that deserves particular protection, as reflected by the WHO guidelines in March 2020 [21]. Regular testing of LTCF employees and residents for COVID-19 was explicitly mentioned by the John Hopkins University in their Guidance on Protecting Individuals Residing in Long-Term Care Facilities [22]. Furthermore, such recommendations can also be found in the WHO policy brief on preventing and managing COVID-19 across long-term care services from July 2020 [38].

To evaluate the effectiveness of testing interventions to protect resident risk groups in LTCFs we extended the model underlying the pandemic preparedness tool CovidSim

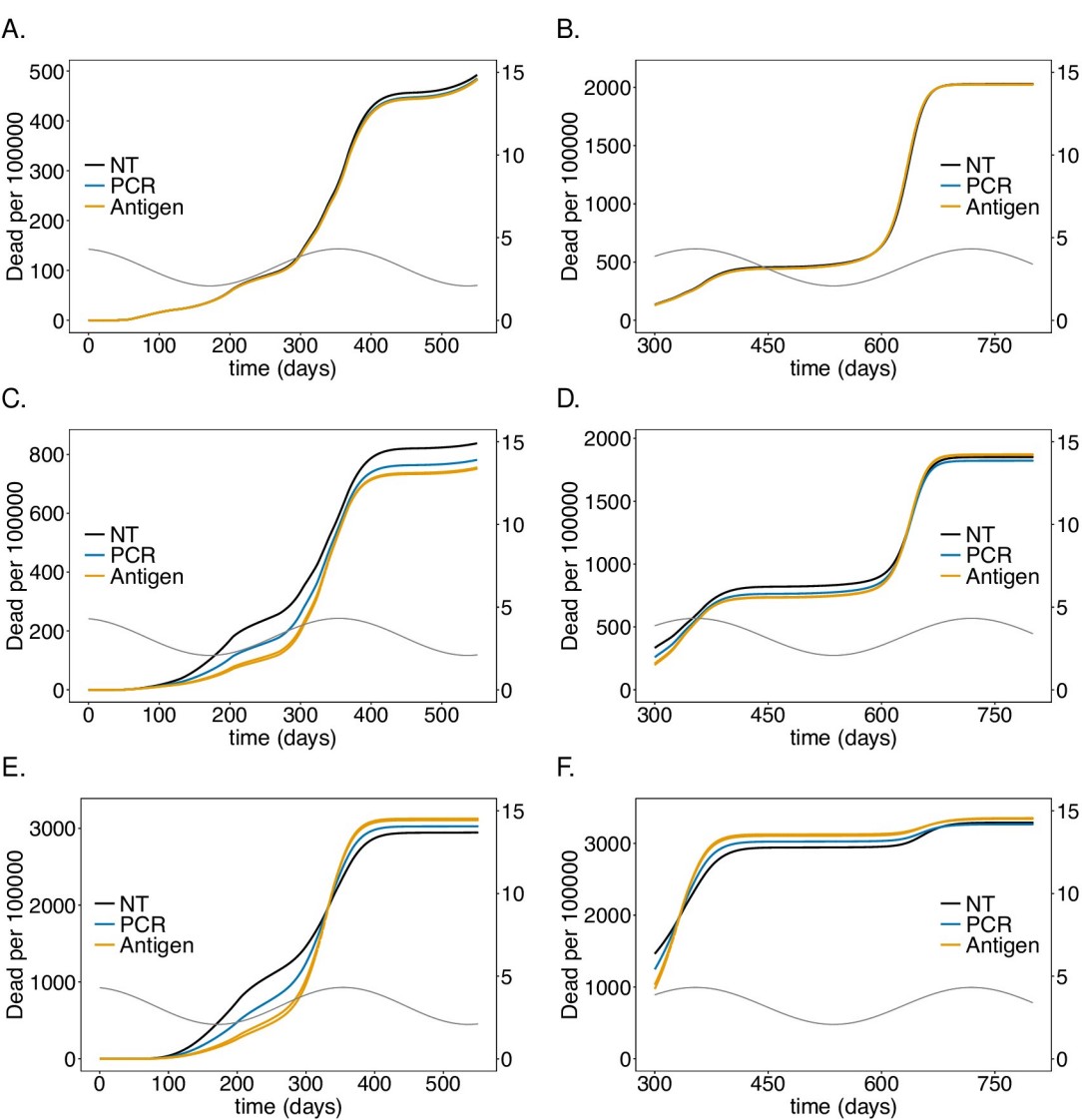

**Fig 11. Impact of PCR *vs.* antigen tests in IFs on mortality: As in Fig 9 but for U.S. IFs instead of German LTCFs.**
Parameters for contact reduction are given in S7 Table.

(http://covidsim.eu; [40]). In particular, the deterministic model formulated as systems of differential equations was extended to separate the risk group of LTCF residents and the LTCF employees from the remaining population. Control interventions within the LTCFs affecting the residents and employees roughly reflect the WHO recommendations [38]. Besides, LTCF employees are regularly tested for COVID-19 and isolated from the resident risk groups if the result is positive. In addition to these control interventions general contact isolation measures (reducing effectively the number of contacts of each individual) and case isolation measures (quarantine and home isolation) were implemented more flexibly than in CovidSim. Importantly, general contact reduction affects residents and employees in LTCFs differently than the remaining population. Especially, general contact reduction measures between residents of LTCFs are difficult to enforce.

In our investigations, the model was adjusted to reflect the situation in the Federal Republic of Germany. However, the model is not restricted to one particular country but will be applicable to any other industrialised nation with a similar age structure—namely one for which age-dependent contact behaviors can be neglected (extending the model to include age dependence is possible in principle). The model per se is not restricted to the situation of LTCFs but applies similarly to other closed facilities. To illustrate this, we parameterized the model to reflect the situation of incarceration facilities (IF) in the United States of America. Inmates in U.S. prisons share several risk factors with LTCF residents (pre-existing conditions etc.) and have an increasing elderly population. U.S IFs are of particular interest since the number of COVID-19 within them was reported to be extremely high [56].

The results clearly indicate that regular COVID-19-screening of LTCF employees by testing successfully reduces the number of cases and deaths in the resident risk group and among LTCF employees. However, even with thorough testing a reduction of only 50% of cases can be achieved and only in combination with strong distancing measures that control the epidemic in the overall population. Our results indicate that the frequency at which employees are tested has a strong effect. Testing once every working week, i.e., every 5 days with good quality PCR tests is less effective than testing daily with good quality antigen rapid tests—testing more frequently would improve the outcome but seems unrealistic. The waiting time for the return of test results (ranging from 12 to 96 hours) has a less noticeable effect. A waiting time of 24-48 hours for PCR tests and 15 minutes for antigen tests seems to be realistic. The quality of the test in terms of sensitivity has a clear impact on the outcome. Here, PCR tests were assumed to be relatively conservative, considering the fact that these tests are constantly improved. Our simple rough estimates of the economic gain of the proposed intervention, comparing the potential costs of COVID-19 treatments with the costs for the testing intervention are less encouraging. However, these estimates are conservative as they do not account for healthcare costs for long-term effects of the infection and capacity shortages in the LTCFs, e.g., due to isolation measures of infected residents. Notably, the estimated costs are more encouraging for antigen rapid tests. (Note that the model is only applicable to highly specific antigen-tests for which false-positive results can be ignored.) Moreover, antigen tests have a lower sensitivity in the early stages of the infection, when antigen levels in the patient have not yet reached the detection threshold, therefore more frequent testing is required. However, COVID-19 rapid tests are cost efficient, with approximate 3-10 Euro per test. With ongoing vaccination campaigns targeting risk groups first, the economic considerations and the effectiveness of testing intervention need to be reevaluated. Altogether, our results clearly indicate that testing alone is insufficient to protect the risk group. In particular, our results highlight the danger of the controversial strategies aiming for herd immunity, while protecting only vulnerable risk groups as initially advocated by, e.g., the U.K. government [58].

The proposed intervention considers regular testing only of LTCFs employees (staff) not of residents. The reason is that we wanted to study the impact of minimal-invasive control measure. Namely, the risk group is twice as large as the target population being tested. Both, the additional costs and discomfort of testing for elderly people would be substantial. Our model covers the additional requirement of visitors (from the general population) to present negative COVID-19 tests upon entering LTCFs. This was ignored in the simulations to obtain conservative predictions.

It should be noted that general contact reduction measures are modelled time-dependently here. While our example assumed intuitive reductions that reflect the imposed interventions at the beginning of the pandemic, we assumed rather naive reductions for the future. In particular, no contact reductions after summer 2021 were assumed in the general population. These assumptions were made to study the effect of testing and the model behavior in an

uncontrolled epidemic scenario. With ongoing vaccination campaigns such a hypothetical scenario is too extreme. However, model parameters can be adapted to reflect more realistic situations.

Besides a number of similarities, our simulations suggest that the implications of testing staff are very different for U.S. IFs. This is mainly due to the difference in nature of the contact behavior in U.S. IFs and German LTCFs. Particularly, U.S. prisons are typically highly crowded and prisoners have many close contacts (in sleeping areas, dining facilities, showers, etc.) [55, 56]. Precautionary measures such as systematic use of filtering face-piece respirators (FFR) are impossible to enforce in an efficient way. Namely, FFRs cannot be worn for more than 6-8 hours without proper reprocessing [59, 60]. Therefore, unlike LTCF residents, incarcerated persons (i) have on average more contacts that could potentially lead to disease transmission than the general population, and (ii) are less protected by contact-reducing measures than the general population. (Note, not every encounter between two individuals would qualify as a "contact" in the model, because exposure time is too short.) Due to these facts the epidemic leads to higher incidence in IFs than in the remaining population. Introducing COVID-19 into an IF at the beginning of the epidemic is limited and might be controllable. However, once COVID-19 spreads into the IF, it is impossible to contain under crowded conditions. Hence, initially, incidence in IFs lacks behind that in the general population and testing contributes to efficiently reduce the number of infections. However, this preserves the susceptible population in IFs. Consequently, in the presence of seasonal fluctuations in $R_0$, the epidemic can gain more momentum, leading to an overall higher number of infections. Hence, testing is counterproductive. In fact, our simulations suggest, that testing would have lead to an even higher number of infections during the epidemic peak in the holiday season 2020. Moreover, our model suggests that levels of immunity, which accumulated during the past epidemic peaks, are sufficiently high to render future epidemic outbreaks less severe. Importantly, such implications must be taken with caution, because the herd immunity argument as implied by the model neglects mutations in the virus and assumes recovered individuals to be permanently immune. However, immunity might be only temporal and even if it was permanent, recovered individuals might be infected again with novel viral strains [61, 62].

The model predictions here are based on assumptions such as permanent immunity after recovery from infections and assumes only one viral variant. The assumption of immunity is appropriate for a short period of time (6 months to one year), but it is unclear how restrictive this assumption is on the long run. Mutant variants such as the British or South African variants are characterized by higher base reproductive numbers [63]. This can be incorporated in the model approximately by increasing the annual average base reproductive number in a sigmoidal fashion, synchronized with the spread of these variants. However, this is not considered in the results presented here. Moreover, the model does not consider vaccination campaigns. These already have a substantial effect on mortality in LTCFs [64]. Including vaccination campaigns would imply substantial model extensions that can be ascertained from the vaccination model of [65].

Summarizing, the predictive model introduced here suggests that testing employees in closed facilities is only efficient in combination with thorough epidemic management, and particular contact reductions. The less controlled the epidemic, the less effective is testing. As suggested by empirical evidence [55, 56], the contact behavior in U.S. prisons hampers efficient implementation of preventative measures and indicates that the U.S correctional system is/was not prepared for a global pandemic. Prioritizing IFs residents during vaccination campaigns might relieve the critical situation in these facilities, especially because of high numbers of preexisting healthcare conditions among prisoners [37]. To refine predictions the readers

are encouraged to use the accompanying Python code (available at https://github.com/Maths-against-Malaria/COVID19_LTCFs.git).

## Supporting information

**S1 Fig. Impact of rate of testing IF staff on the number of infections: As in Fig 2 but for U.S.** IFs instead of German LTCFs. Parameters for contact reduction are given in S7 Table.
(PDF)

**S2 Fig. Impact of rate of testing IF staff on on mortality: As in Fig 3 but for U.S. IFs instead of German LTCFs.** Parameters for contact reduction are given in S7 Table.
(PDF)

**S3 Fig. Impact of processing time of testing IF staff on the number of infections: As in Fig 4 but for U.S.** IFs instead of German LTCFs. Parameters for contact reduction are given in S7 Table.
(PDF)

**S4 Fig. Impact of processing time of testing IF staff on on mortality: As in Fig 5 but for U.S.** IFs instead of German LTCFs. Parameters for contact reduction are given in S7 Table.
(PDF)

**S5 Fig. Impact of sensitivity when testing IF staff on the number of infections: As in Fig 6 but for U.S.** IFs instead of German LTCFs. Parameters for contact reduction are given in S7 Table.
(PDF)

**S6 Fig. Impact of sensitivity when testing IF staff on mortality: As in Fig 7 but for U.S.** IFs instead of German LTCFs. Parameters for contact reduction are given in S7 Table.
(PDF)

**S1 Appendix. Mathematical description.**
(PDF)

**S1 Table. (Sub-) population sizes of Germany (GER) and the USA chosen in simulations.**
(PDF)

**S2 Table. Summary of model parameters and their choices for numerical simulations for Germany (GER) and USA.**
(PDF)

**S3 Table. Summary of variables describing sub-population sizes in non- infectious (sub-) states in Germany (GER) and the USA.**
(PDF)

**S4 Table. Summary of variables describing sub-population sizes in infectious (sub-)states in Germany (GER) and the USA.**
(PDF)

**S5 Table. Summary of model parameters describing the contact behavior for Germany (GER) and USA.**
(PDF)

**S6 Table. Contact reduction parameters chosen for the simulations of Germany.**
(PDF)

**S7 Table. Contact reduction parameters chosen for the simulations of USA.**
(PDF)

## Acknowledgments

We want to dedicate this work to all the victims of the SARS-CoV-2 virus. Our grief is with the friends and families of the dreadful disease. The authors like to express their sympathy to all working to find a cure for the virus. The fruitful discussions with Prof. Martin Eichner and his helpful comments and inputs are gratefully acknowledged. The authors are grateful to two anonymous reviewers and the editors for very constructive comments that helped to improve the manuscript.

## Author Contributions

**Conceptualization:** Henri Christian Junior Tsoungui Obama, Nessma Adil Mahmoud Yousif, Looli Alawam Nemer, Pierre Marie Ngougoue Ngougoue, Gideon Akumah Ngwa, Miranda Teboh-Ewungkem, Kristan Alexander Schneider.

**Formal analysis:** Henri Christian Junior Tsoungui Obama, Nessma Adil Mahmoud Yousif, Looli Alawam Nemer, Pierre Marie Ngougoue Ngougoue, Kristan Alexander Schneider.

**Funding acquisition:** Kristan Alexander Schneider.

**Investigation:** Henri Christian Junior Tsoungui Obama, Nessma Adil Mahmoud Yousif, Looli Alawam Nemer, Pierre Marie Ngougoue Ngougoue, Kristan Alexander Schneider.

**Methodology:** Henri Christian Junior Tsoungui Obama, Nessma Adil Mahmoud Yousif, Looli Alawam Nemer, Pierre Marie Ngougoue Ngougoue, Kristan Alexander Schneider.

**Software:** Henri Christian Junior Tsoungui Obama, Kristan Alexander Schneider.

**Supervision:** Miranda Teboh-Ewungkem, Kristan Alexander Schneider.

**Validation:** Kristan Alexander Schneider.

**Visualization:** Henri Christian Junior Tsoungui Obama, Looli Alawam Nemer, Kristan Alexander Schneider.

**Writing – original draft:** Henri Christian Junior Tsoungui Obama, Nessma Adil Mahmoud Yousif, Looli Alawam Nemer, Pierre Marie Ngougoue Ngougoue, Kristan Alexander Schneider.

**Writing – review & editing:** Henri Christian Junior Tsoungui Obama, Nessma Adil Mahmoud Yousif, Looli Alawam Nemer, Pierre Marie Ngougoue Ngougoue, Gideon Akumah Ngwa, Miranda Teboh-Ewungkem, Kristan Alexander Schneider.

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
