## [Decision Letter · Decision Letter 0]

23 Nov 2020

PONE-D-20-31748

Preventing COVID-19 spread in closed facilities by regular testing of employees - an efficient intervention in long-term care facilities and prisons

PLOS ONE

Dear Dr. Schneider,

Thank you for submitting your manuscript to PLOS ONE. I have now received recommendations and comments from reviewers. From their reports, the reviewers have appreciated your work and recommended it for the following stage of the publication path. Therefore, I invite you to submit a revised version of the manuscript that addresses the points raised during the review process.

The revised version should:

Carefully consider all comments raised by reviewers and describe how each one of them has been addressed. Particular care in the answer is required in case the authors decide not to address a reviewer's comment. Motivations should be explained plainly. In the same way, consider the comments that I added in the following "Additional Editor Comments" section. 

As a general comment, I share the reviewers opinion that yours is a scientific work that has clear merits and presents valuable results, and also satisfies PLOS ONE publication criteria. The contribution is, however, not always sufficiently explicit and easy to understand. Additional efforts to make it clearer will contribute to let the work be appreciated by the readers and the scientific community.  

We look forward to receiving your revised manuscript.

Kind regards,

Marco Cremonini, Ph.D.

Academic Editor

PLOS ONE

Journal Requirements:

2. Please consider whether the word 'draconic' could be substituted for a less subjective term.

Additional Editor Comments :

Results

line 173: Parameters for R0 should be presented with context and motivations, in particular the assumption about the seasonal variation needs references and a discussion, since it is important for the model and for the results, given that the root causes of Covid-19 seasonality are still debated.

Figures showing subplots (i.e., Fig.2 and following ones), are not easy to read, being dense, and visual comparisons between plots are typically difficult. A simplification and different organization of the results could be worthwhile, to make results easier to understand.

For instance, the epidemic classes (S, I, R, and D) are not independent one to the other, therefore not all of the corresponding plots are strictly needed, at least as a visual representation. For instance, plots of infections are necessary, not so to always show all the others, from Fig 2 to Fig 5.

Also, beside visual representations, tables could be effective too, especially to ease comparisons between tests.

For example, it is not immediately clear what the important information is by looking at S, R, and D panels. Similarly, it is not immediately clear how infections are changing among the four cases.

For these reasons, I suggest to consider to reorganize the presentation of these results, with one that more explicitly highlight the key contributions of this work.

Reviewers' comments:

Reviewer's Responses to Questions

**Comments to the Author**

1. Is the manuscript technically sound, and do the data support the conclusions?

Reviewer #1: Yes

Reviewer #2: Partly

2. Has the statistical analysis been performed appropriately and rigorously? 

Reviewer #1: I Don't Know

Reviewer #2: N/A

3. Have the authors made all data underlying the findings in their manuscript fully available?

Reviewer #1: Yes

Reviewer #2: Yes

4. Is the manuscript presented in an intelligible fashion and written in standard English?

Reviewer #1: Yes

Reviewer #2: Yes

5. Review Comments to the Author

Reviewer #1: This is a well-written and well-organized manuscript presenting a relatively clear model analyzing the impact of different intervals of testing on the number of infections and deaths among residents in long-term care facilities. I am not in expert in the underlying mathematical model used to predict these effects; I read this instead as a scholar of closed institutions, especially prisons, and focus on the theoretical framing and implications of the results. I trust other reviewers will speak more directly to the underlying mathematical model. In terms of how these findings are presented, I have two central suggestions: one about the theoretical framing, and one about the way the mathematically modeled impacts are highlighted throughout the piece. Please see attached for a discussion of each.

Reviewer #2: The paper deals with a very important hot topic. It is well organized, clear in the rationale and appropriate in the choice of the different variables of the simulation. My doubts concern the economic parts. Even if the authors said that they did a rough estimation of economic costs, it is not clear whether the value chosen are related to Germany or refer to other official sources of information (such as Diagnosis related groups) and what these costs comprise: drug, medical staff, equipment…? Within a LTCF facility? My concern is also that authors, even if indirectly, relate flue costs with the COVID ones. I’m not sure this is correct. My suggestion is to detail this part of the analysis, better explaining their estimation.

6. PLOS authors have the option to publish the peer review history of their article (what does this mean?). If published, this will include your full peer review and any attached files.

Reviewer #1: No

Reviewer #2: No

---

## [Decision Letter · Decision Letter 1]

16 Mar 2021

PONE-D-20-31748R1

Preventing COVID-19 spread in closed facilities by regular testing of employees - an efficient intervention in long-term care facilities and prisons?

PLOS ONE

Dear Dr. Schneider,

Thank you for submitting your manuscript to PLOS ONE. The reviewers have both appreciated your work and few details are left to revise. Please, carefully consider the comments of Reviewer 1, which are aimed at better clarify some aspects that merit special attention from the readers, and submit a revised version of the manuscript that addresses the points raised during the review process.

We look forward to receiving your revised manuscript.

Kind regards,

Marco Cremonini, Ph.D.

Academic Editor

PLOS ONE

Journal Requirements:

Reviewers' comments:

Reviewer's Responses to Questions

**Comments to the Author**

1. If the authors have adequately addressed your comments raised in a previous round of review and you feel that this manuscript is now acceptable for publication, you may indicate that here to bypass the “Comments to the Author” section, enter your conflict of interest statement in the “Confidential to Editor” section, and submit your "Accept" recommendation.

Reviewer #1: (No Response)

Reviewer #2: All comments have been addressed

2. Is the manuscript technically sound, and do the data support the conclusions?

Reviewer #1: Yes

Reviewer #2: Yes

3. Has the statistical analysis been performed appropriately and rigorously? 

Reviewer #1: Yes

Reviewer #2: N/A

4. Have the authors made all data underlying the findings in their manuscript fully available?

Reviewer #1: Yes

Reviewer #2: No

5. Is the manuscript presented in an intelligible fashion and written in standard English?

Reviewer #1: Yes

Reviewer #2: Yes

6. Review Comments to the Author

Reviewer #1: Overall, I thought this paper constituted an engaged, responsive, thoughtful revision from the earlier draft. I had a just a few minor thoughts about further refinement to clarify the contribution: 1. Although the authors do a much better job in this draft explaining and comparing their selection of LTCFs in Germany and IFs in the U.S., there are still a few places where these comparisons could be clarified, especially (a) in the introduction where the phrase "a similar reasoning applies" could be made more precise to introduce and specify the exact analogy between LTCFs and IFs and (b) in the IFs findings section, more explicit connections could be made to each of the LTCF sub-sections of test processing, test sensitivity, antigens, and economic analysis (this last seemed like a particularly weird absence in the IF section). 2. The economic analysis was intriguing, but it seems worth adding at least a sentence noting all the things that cannot quite be quantified (the value of a life saved, the potential costs of long-term effects of having been infected, etc.) 3. In the IFs findings section, I remained a bit confused, even after reading a few times, about the exact effects of testing on IF infection rates. Could the authors state the finding a bit more clearly with explicit comparisons/differences with the LTCF model (this partly relates to point 1(b)). 4. Are there things this model does not account for that might change the outcomes? In particular, we do not yet know the effects of variants. Might the authors devote a sentence or two to specify potential limitations, like lack of knowledge about variants or long-time viral effects, to their model? 5. Just a very minor point: the authors use "anyhow" as a transitional phrase in a few places and it feels a bit overly casual/colloquial.

Reviewer #2: The manuscript has been revised and updated addressing reviewers’ comments and suggestions. The final version has been highly improved and can be accepted for publication.

7. PLOS authors have the option to publish the peer review history of their article (what does this mean?). If published, this will include your full peer review and any attached files.

Reviewer #1: No

Reviewer #2: No

---

## [Editor Report · Decision Letter 2]

22 Mar 2021

Preventing COVID-19 spread in closed facilities by regular testing of employees - an efficient intervention in long-term care facilities and prisons?

PONE-D-20-31748R2

Dear Dr. Schneider,

We’re pleased to inform you that your manuscript has been judged scientifically suitable for publication and will be formally accepted for publication once it meets all outstanding technical requirements.

Kind regards,

Marco Cremonini, Ph.D.

Academic Editor

PLOS ONE
---

## [Editor Report · Acceptance letter]

8 Apr 2021

PONE-D-20-31748R2 

Preventing COVID-19 spread in closed facilities by regular testing of employees – an efficient intervention in long-term care facilities and prisons? 

Dear Dr. Schneider:

I'm pleased to inform you that your manuscript has been deemed suitable for publication in PLOS ONE. Congratulations! Your manuscript is now with our production department. 

Kind regards, 

on behalf of

Dr. Marco Cremonini 

Academic Editor

PLOS ONE